# Protective Attitudes toward Occupational Radiation Exposure among Spine Surgeons in Japan: An Epidemiological Description from the Survey by the Society for Minimally Invasive Spinal Treatment

**DOI:** 10.3390/medicina59030545

**Published:** 2023-03-10

**Authors:** Yasukazu Hijikata, Yoshihisa Kotani, Akinobu Suzuki, Koichi Morota, Haruki Funao, Masayuki Miyagi, Tadatsugu Morimoto, Haruo Kanno, Ken Ishii

**Affiliations:** 1Spine and Low Back Pain Center, Kitasuma Hospital, Kyoto 654-0102, Japan; 2Department of Orthopedic Surgery, Kansai Medical University Medical Center, Osaka 570-8507, Japan; 3Department of Orthopaedic Surgery, Osaka Metropolitan University, Osaka 545-8585, Japan; 4Department of Radiology, Shinkomonji Hospital, Fukuoka 800-0057, Japan; 5Department of Orthopaedic Surgery, School of Medicine, International University of Health and Welfare, Chiba 286-0048, Japan; 6Department of Orthopedic Surgery, School of Medicine, Kitasato University, Sagamihara 252-0375, Japan; 7Department of Orthopaedic Surgery, Faculty of Medicine, Saga University, Saga 849-8501, Japan; 8Department of Orthopaedic Surgery, Tohoku Medical and Pharmaceutical University, Sendai 983-8536, Japan; 9Department of Orthopaedic Surgery, Keio University School of Medicine, Tokyo 160-8582, Japan; 10Society for Minimally Invasive Spinal Treatment, Tokyo 101-0063, Japan

**Keywords:** minimally invasive, spine, questionnaire survey, ionizing radiation, protection, occupational radiation exposure, behavioral change stage, trans theoretical model

## Abstract

*Background and Objectives*: The global trend toward increased protection of medical personnel from occupational radiation exposure requires efforts to promote protection from radiation on a societal scale. To develop effective educational programs to promote radiation protection, we clarify the actual status and stage of behavioral changes of spine surgeons regarding radiation protection. *Materials and Methods*: We used a web-based questionnaire to collect information on the actual status of radiation protection and stages of behavioral change according to the transtheoretical model. The survey was administered to all members of the Society for Minimally Invasive Spinal Treatment from 5 October to 5 November 2020. *Results*: Of 324 members of the Society for Minimally Invasive Spinal Treatment, 229 (70.7%) responded. A total of 217 participants were analyzed, excluding 12 respondents who were not exposed to radiation in daily practice. A trunk lead protector was used by 215 (99%) participants, while 113 (53%) preferred an apron-type protector. Dosimeters, thyroid protector, lead glasses, and lead gloves were used by 108 (50%), 116 (53%), 82 (38%), and 64 (29%) participants, respectively. While 202 (93%) participants avoided continuous irradiation, only 120 (55%) were aware of the source of the radiation when determining their position in the room. Regarding the behavioral change stage of radiation protection, 134 (62%) participants were in the action stage, while 37 (17%) had not even reached the contemplation stage. *Conclusions*: We found that even among the members of the Society for Minimally Invasive Spinal Treatment, protection of all vulnerable body parts was not fully implemented. Thus, development of educational programs that cover the familiar risks of occupational radiation exposure, basic protection methods in the operating room, and the effects of such protection methods on reducing radiation exposure in actual clinical practice is warranted.

## 1. Introduction

Recent advances in surgical techniques and equipment have led to the development of minimally invasive spine stabilization (MISt) in the field of spine surgery [1,2,3]. While MISt procedures can reduce patient burden, there is growing concern about the associated potential for increased radiation exposure for surgeons and other medical personnel. It has been reported that exposure to radiation during MISt procedures is within permissible limits when adequate radiation protection is used [4]. However, it should be noted that in the case of surgeons who are indifferent to protection from radiation, the dose of radiation exposure is unknown, and the impact on health cannot be assessed.

Regarding the effects of occupational radiation exposure on medical personnel, there are concerns about the effects of direct exposure to the hands [5,6,7,8,9], and indirect exposure, exposure due to scattered radiation, to radiosensitive organs such as the lens of the eye, thyroid gland, and reproductive organs [10,11,12]. These effects are caused by non-uniform exposure and cannot be prevented with a trunk lead protector alone. In recent years, efforts have been made to promote protection against occupational radiation exposure among medical personnel worldwide [13]. This entails efforts to promote radiation protection on a societal scale.

To promote radiation protection, it is necessary to develop effective educational programs to promote behavioral changes in the target population. The first step for developing such educational programs is collecting information on the actual status of radiation protection and readiness of the target population to change their behavior concerning radiation protection use. In the past few years, the results of surveys vis à vis radiation protection awareness and use by orthopedic surgeons have been reported from various countries [14,15,16], and it has become clear that protection of all vulnerable body parts is not sufficiently implemented. However, there has not been enough research on the readiness to change the behavior toward radiation protection. Furthermore, there are limited reports on the actual status of protection from occupational exposure among spine surgeons in Japan.

Hence, to develop educational programs on radiation protection for spine surgeons, we first conducted a survey of members of the Society for Minimally Invasive Spinal Treatment (MIST) regarding the actual status and stages of behavioral change for radiation protection.

## 2. Materials and Methods

### 2.1. Study Population

This study is a cross-sectional survey for the Society for MIST in Japan. The Society for MIST is engaged for propagation of safe MIST and consists mainly of surgeons who are routinely involved in spine surgery. The survey was administered using a web-based questionnaire between 5 October and 5 November 2020, which was sent to all members of the Society for MIST who had a valid e-mail address on the date of the survey. Those who refused to participate and those who did not use X-rays in their practice at the time of the survey were excluded. This study was approved by relevant certified institutional review boards, and all participants provided informed consent electronically.

### 2.2. Data Collection

We used a web-based questionnaire to collect information on the background characteristics, actual status of radiation protection, and stage of behavioral change regarding radiation protection. Background characteristics included information on age, sex, total years of experience, and frequency of radiation use—the average number of days per week that radiation is used. Regarding the actual status of radiation protection, participants were asked whether they used a trunk lead protector, dosimeters, lead glasses, thyroid protector, and lead gloves. The type of trunk protector used—apron type, coat type, or skirt type—was also determined. In addition, we questioned whether participants were aware of the radiation source and positioned themselves accordingly, and whether they avoided continuous exposure. 

Regarding the stages of behavioral change concerning radiation protection, participants were asked to pick one of the following choices about “efforts toward radiation protection in addition to trunk lead protectors and dosimeters”: (1) Not necessary, (2) it may be necessary, but don’t care about it, (3) intend to act within 6 months, (4) intend to act within a month, (5) started to work within 6 months, (6) have been working for longer than 6 months. Each answer was classified according to the transtheoretical model [17], with (1) and (2) corresponding to the precontemplation stage, (3) to the contemplation stage, (4) to the preparation stage, (5) to the action stage, and (6) to the maintenance stage.

### 2.3. Statistical Analysis

Values were expressed as the median and interquartile range (IQR) for continuous variables and absolute and relative frequencies for dichotomous or categorical variables. First, the background characteristics of the participants were described. Second, the actual status and the stage of behavioral change regarding radiation protection were described. Further, the actual status of radiation protection for subgroups of participants who reached the action stage were described. All statistical analyses were performed using Stata (version 17.0, StataCorp LLC, College Station, TX, USA).

## 3. Results

A web-based questionnaire survey was conducted among 324 members of the Society for MIST, and 229 (70.7%) responded. After excluding 12 participants who were not currently involved in medical treatment using radiation, a total of 217 participants were analyzed. Table 1 summarizes the background characteristics of the participants. The median age was 46 years (IQR: 42–50), the majority of participants were male (99%), and the median number of work years was 20 years (IQR: 16–25). One hundred and thirty-two participants (61%) worked in practices that used radiation three or more days per week.

The actual status of radiation protection is depicted in Table 2. Trunk protectors were used by 215 participants (99%), and many of them used apron-type protectors. Dosimeters and thyroid protectors were used by about half of the participants, and lead glasses and lead gloves were used by about one-third. Of those who used lead gloves, 26 (41%) used only lead gloves, not in combination with lead glasses. More than 90% of the participants avoided continuous irradiation, but only about half of them were aware of the radiation source and accordingly decided where to stand. The stage of behavioral change of radiation protection is also presented in Table 2. One hundred and thirty-four (62%) participants were in the action stage, while thirty-seven (17%) had not even reached the contemplation stage. Among participants who were in the action stage, a dosimeter, lead glasses, and lead gloves were used by about half (Table 3). 

## 4. Discussion

A web-based questionnaire survey of 229 (70.7%) members of the Society for MIST was conducted to obtain information on the actual status of radiation protection and the stages of behavioral change concerning radiation protection. As a result, it was found that even among the members of the Society for MIST, the protection of all vulnerable body parts was not fully implemented, and about 17% of the members considered radiation protection other than trunk lead protectors and dosimeters unnecessary. Radiation protection may not be a critical issue for surgeons who use advanced technologies such as robot-assisted systems and rarely have the opportunity to use radiation. However, several other surgeons would benefit from educational programs promoting radiation protection, and the information on the actual status and stages of behavioral change of radiation protection identified in this study will be valuable for the development of such educational programs.

### 4.1. Comparison to Previous Studies

Falavinga et al. conducted a survey of 1450 AO Spine Latin America Society members in December 2016 (response rate 25.6%) and reported that the percentage of use of protective devices was 20% for lead glasses, 64% for thyroid protectors, and 7% for lead gloves [13]. Fidan et al. conducted a survey of 323 orthopedic surgeons in Turkey (response rate 86%) in 2019 and reported that about 30% of the respondents were not routinely involved in radiation protection, 68% used a lead apron, 11% used dosimeters regularly, 1.7% used lead glasses, 52% used a thyroid protector, and 0.5% used lead gloves [14]. Kang et al. conducted a survey at the 2017 Korean Orthopedic Association conference (513 respondents) and found that 52% of the respondents always used a lead apron, 29% regularly used dosimeters, 3.5% always used lead glasses, 29% always used a thyroid protector, and 2.5% always used lead gloves [15]. Direct comparisons cannot be made between the participants of the previous studies and those of the present study because of differences in countries, environments, and survey periods. Nevertheless, the Society for MIST might be more advanced in radiation protection, especially in terms of the use of lead glasses (38%) and lead gloves (29%). This may be related to the fact that the Society for MIST is a highly specialized society. However, the proportion of use of lead glasses, which is particularly important for protection against exposure to the lens of the eye, and application of awareness of the radiation source to help decide where to stand while conducting the procedure need to be improved.

### 4.2. Exposure to the Lens of the Eye

Conventional recommendations regarding the permissible limits for the eye lens were based on the assumption that micro-opacities do not necessarily progress to vision-impairing cataracts. However, in recent years, this assumption has been overturned, and in 2012, the International Commission on Radiological Protection issued a recommendation to drastically lower the permissible exposure limit for the eye lens [18]. Considering the recommendation, the permissible exposure limits for the eyes of workers who use radiation are being lowered in Europe and other countries around the world. The Regulation for Prevention of Ionizing Radiation Hazards (Law number: Ministry of Labour Order No. 41 of 1972) in Japan was revised in April 2021; a limit of lens equivalent dose of 100 mSv/5 years and no more than 50 mSv/year in any year was established. With appropriate exposure protection, lens exposure in spine surgery has been reported to be within the permissible range (0.07 mSv per spinal fusion procedure) [4]. However, it should be noted that without appropriate protection, radiation exposure may exceed the permissible limit. Nagamoto et al. conducted a survey of physicians engaged in medical treatment using radiation in angiography or endoscopy rooms and suggested that there may be a certain number of physicians whose equivalent dose of the lens exceeds 20 mSv/year; for instance, in 2018, 4 out of 15 physicians exceeded 20 mSv/year and 3 did not even use a dosimeter [19]. Exposure to the lens of the eye is mainly due to scattered radiation from the patient. To protect the lens from exposure to scattered radiation, it is necessary to know that the exposure dose varies greatly depending on the position where the physician stands [20], and to use lead glasses, which have been proven to be effective in protecting the lens [21]. We believe that the compliance rate of these radiation protection practices needs to be as close to 100% as possible.

### 4.3. Exposure to the Skin of the Hands

Orthopedic and spine surgeons should also be aware of the direct exposure of their hands. It has been reported that the radiation dose to the hand is well within the permissible range if the hand is not held directly over the irradiation field (0.32 mSv per spinal fusion procedure compared to the skin equivalent dose limit of 500 mSv/year) [4]. However, most surgeons experience the reflection of their own hand on the monitor projecting the surgical field in clinical practice. In our previous study, 227 out of 229 participants had experienced direct exposure to the hand; more than 60% reported “frequently” or “almost always” in terms of the intensity, and three reported skin cancer [9]. It is difficult to measure the direct radiation to the hands, and even the level of exposure in daily practice is unknown. In recent years, the relationship between direct radiation exposure to the hands and various health risks has been attracting attention [5,6,7,8,9], and it is necessary to promote protection from direct radiation exposure to the hands in addition to protection against indirect exposure to the lens of the eye. Lead gloves may not be necessary if the best direct exposure protection, “keeping one’s own hands out of the irradiation field,” is adhered to. In addition, lead gloves do not provide adequate protection against direct radiation exposure because they are designed for protection against scattered radiation. However, considering that many surgeons actually place their own hands in the irradiation field, the 29% use rate of lead gloves in the present study may not be sufficient. Furthermore, it should be noted that 41% of the participants who used lead gloves in the present study did not use lead glasses. The use of lead gloves alone may lead to indirect exposure to the lens of the eye due to scattered radiation generated by the gloves, and it is desirable to use them in combination with lead glasses.

### 4.4. Evidence Needed to Develop Educational Programs

The behavioral change stage of the Society for MIST members predominantly comprised precontemplation and contemplation (17% and 16%, respectively). Therefore, educational programs targeting physicians at those behavioral change stages are desirable. An appropriate intervention for behavioral change of participants in the precontemplation stage is educating them about familiar risks and basic preventive behaviors [17]. To change the behavior of participants in the contemplation stage, it is useful to provide information regarding the benefits of preventive behavior, which is essentially the extent to which exposure can be reduced by actual protection behavior [17]. Therefore, dispensing information on health risks due to radiation exposure such as occurrence of cataracts, dermatitis, and skin cancer; actual radiation doses in daily practice; and evidence regarding appropriate protection against occupational radiation exposure and its actual dose reduction effects in operating rooms is required. A systematic review of evidence on these areas is warranted, and additional studies should be performed on areas where the evidence is insufficient. This will enable us to develop effective educational materials for radiation protection, which could be used as educational programs in the form of short videos and lectures. This, in turn, will help to promote safe spine surgery.

### 4.5. Study Limitations

There are several limitations in this study. The first is the timing of the data collection. Since the data collection was conducted before the amendment of the law to lower the lens equivalent dose limit (April 2021), the behavioral change stage of radiation protection may have possibly progressed due to the amendment of the law. The second is misclassification. Those who responded, “Not necessary” (17%) may include those who rarely use X-rays due to the use of navigation or robotics; if X-rays are not used, protection would be no longer necessary; this is the best form of radiation protection. However, radiation protection is still necessary for procedures such as myelography and nerve root blocks, which also use X-rays. The other limitation is generalizability. As can be seen from the comparison with other countries, the Society for MIST comprises physicians who are highly aware of radiation protection, and the results might not be generalizable. Moreover, the fact that the survey was conducted by a Japanese academic society and that most of the participants were Japanese people living in Japan also limits the generalizability.

## 5. Conclusions

In the present study, we found that even among the members of the Society for MIST, the protection of the vulnerable body parts against occupational radiation exposure was not fully implemented. In addition, we found that 17% of participants had not reached the contemplation stage of radiation protection. There is a need to develop educational programs that cover the familiar risks, basic protection methods, and the actual effects of protection methods against occupational radiation exposure.

## Figures and Tables

**Table 1 medicina-59-00545-t001:** Background characteristic of the participants.

	Total
	*n* = 217
Age (years)	46 (42–50)
Male sex	215 (99)
Years of experience	20 (16–25)
Frequency of radiation use, days per week	
1 day	26 (12)
2 days	59 (27)
3 days	82 (38)
4 days	34 (16)
5 days	16 (7)

Data were number (%) and median (interquartile range).

**Table 2 medicina-59-00545-t002:** Actual status and stage of behavioral change of radiation protection.

	***n* = 217**
Actual status of radiation protection	
Trunk protector	215 (99)
Apron type	113 (52) *
Coat type	84 (39) *
Skirt type	18 (8) *
Dosimeter	108 (50)
Lead glasses	82 (38)
Thyroid protector	116 (53)
Lead gloves	64 (29)
Standing aware of the source	120 (55)
Avoid continuous irradiation	202 (93)
Behavioral change stage of radiation protection	
Not necessary/It may be necessary, but don’t care about it (precontemplation)	37 (17)
Intend to act within 6 months (contemplation)	34 (16)
Intend to act within a month (preparation)	12 (6)
Started to work within 6 months (action)	5 (2)
Have been working for longer than 6 months (maintenance)	129 (59)

* Percentage of 215 people who used trunk protector. Data are number (%).

**Table 3 medicina-59-00545-t003:** Actual status of radiation protection of participants who were in the action stage.

	***n* = 134**
Trunk protector	
Apron type	68 (51)
Coat type	57 (43)
Skirt type	9 (6.7)
Dosimeter	77 (57)
Lead glasses	75 (56)
Thyroid protector	97 (72)
Lead gloves	60 (45)
Standing aware of the source	85 (63)
Avoid continuous irradiation	126 (94)

Data are number (%).

## Data Availability

The data presented in this study are available on request from the corresponding author.

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
