# Peer review of "Protective Attitudes toward Occupational Radiation Exposure among Spine Surgeons in Japan: An Epidemiological Description from the Survey by the Society for Minimally Invasive Spinal Treatment"

_medicina, 2023, doi:10.3390/medicina59030545_

Round 1

Reviewer 1 Report

Very interesting topic in spinal surgery with growing interest especially in minimally invasive surgery. 

From our pratical experience, the main target should be to avoid radiation as anyhow possible instead of protective measures.Navigation should further improve lower need of radiation. 

The authors should discuss strategies of how to avoid radiation in spinal surgery to a minimum necessary. 

Author Response

Reviewer 1

Very interesting topic in spinal surgery with growing interest especially in minimally invasive surgery. From our pratical experience, the main target should be to avoid radiation as anyhow possible instead of protective measures. Navigation should further improve lower need of radiation. The authors should discuss strategies of how to avoid radiation in spinal surgery to a minimum necessary.

Response

Thank you for your important remarks. We fully agree with your opinion. The discussion in this study assumes that radiation protection is necessary. However, as you pointed out, the problem would be solved if X-rays were not used. We have refrained from discussing in detail the technology without the use of X-rays because we believe that such a discussion is beyond the scope of this study. Now we understand that it is important, so we have added the following to the Limitation (Page 7, Lines 264-269).

Contents of addition:

The second is misclassification. Those who responded, "Not necessary" (17%) may include those who rarely use X-rays due to the use of navigation or robotics; if X-rays are not used, protection would be no longer necessary and is the best radiation protection. However, radiation protection is still necessary for procedures such as myelography and nerve root blocks, which also use X-rays.

Reviewer 2 Report

I agree that fluoroscopic screw positioning is essential in spine surgery with implants, especially in minimally invasive surgery (MISt).

While surgeons are interested in surgical techniques and their outcomes, they do not focus on radiation protection. I think the content is very valuable in this regard.

The description seems to be simple and the discussion is appropriate. I do not find anything in particular that needs to be revised.

I understand that this study is a survey of the current status of the MISt Society in order to build an educational program, but I was wondering if you could provide an outline of the educational program, if any, that you envisioned.

Author Response

Reviewer  2

I agree that fluoroscopic screw positioning is essential in spine surgery with implants, especially in minimally invasive surgery (MISt). While surgeons are interested in surgical techniques and their outcomes, they do not focus on radiation protection. I think the content is very valuable in this regard. The description seems to be simple and the discussion is appropriate. I do not find anything in particular that needs to be revised. I understand that this study is a survey of the current status of the MISt Society in order to build an educational program, but I was wondering if you could provide an outline of the educational program, if any, that you envisioned.

Response

We appreciate your valuable comments. As you pointed out, the "educational program" was ambiguous in the current situation. We had intended to create educational materials for those in the precontemplation and contemplation stages and to use these materials as educational programs in the form of videos and lectures. In response to your suggestion, we have made the following corrections (Page 7, Lines 257-259).

Before revision:

This will enable us to develop effective educational programs, which in turn will help to promote safe spine surgery.

After revision:

This will enable us to develop effective educational materials for radiation protection, which could be used as educational programs in the form of short videos and lectures. This in turn will help to promote safe spine surgery.